# Surgical Techniques of Gastrojejunostomy in Robotic Pancreatoduodenectomy: Robot-Sewn versus Stapled Gastrojejunostomy Anastomosis

**DOI:** 10.3390/jcm12020732

**Published:** 2023-01-16

**Authors:** Kosei Takagi, Yuzo Umeda, Ryuichi Yoshida, Tomokazu Fuji, Kazuya Yasui, Jiro Kimura, Nanako Hata, Takahito Yagi, Toshiyoshi Fujiwara

**Affiliations:** Department of Gastroenterological Surgery, Okayama University Graduate School of Medicine, Dentistry, and Pharmaceutical Sciences, Okayama 700-8558, Japan

**Keywords:** pancreatoduodenectomy, robotic surgery, gastrojejunostomy, delayed gastric emptying

## Abstract

Background: Delayed gastric emptying (DGE) is a major complication of pancreatoduodenectomy (PD). Several efforts have been made to decrease the incidence of DGE. However, the optimal anastomotic method for gastro/duodenojejunostomy (GJ) remains debatable. Moreover, few studies have reported the impact of GJ surgical techniques on outcomes following robotic pancreatoduodenectomy (RPD). This study aimed to investigate the surgical outcomes of robot-sewn and stapled GJ anastomoses in RPD. Methods: Forty patients who underwent RPD at the Okayama University Hospital between September 2020 and October 2022 were included. The outcomes between robot-sewn and stapled anastomoses were compared. Results: The mean [standard deviation (SD)] operative and GJ time were 428 (63.5) and 34.0 (15.0) minutes, respectively. Postoperative outcomes included an overall incidence of DGE of 15.0%, and the mean postoperative hospital stays were 11.6 (5.3) days in length. The stapled group (n = 21) had significantly shorter GJ time than the robot-sewn group (n = 19) (22.7 min versus 46.5 min, *p* < 0.001). Moreover, stapled GJ cases were significantly associated with a lower incidence of DGE (0% versus 21%, *p* = 0.01). Although not significant, the stapled group tended to have shorter postoperative hospital stays (9.9 days versus 13.5 days, *p* = 0.08). Conclusions: Our findings suggest that stapled GJ anastomosis might decrease anastomotic GJ time and incidence of DGE after RPD. Surgeons should select a suitable method for GJ anastomosis based on their experiences with RPD.

## 1. Introduction

Pancreatoduodenectomy (PD) includes complex reconstructions such as pancreaticojejunostomy, hepaticojejunostomy, and gastrojejunostomy (GJ) [1]. Delayed gastric emptying (DGE), which has an incidence of 20% to 60%, is one of the major complications following PD [2]. Although DGE is not a life-threatening complication, it can cause discomfort to patients and reduce the quality of life, leading to prolonged hospital stays [3]. With regard to the optimal anastomotic method for GJ, controversy still exists on which anastomotic method is the best in open PD: stapled versus hand-sewn anastomosis, and GJ versus duodenojejunostomy [4]. Moreover, to date, few studies have reported on surgical outcomes comparing robot-sewn versus stapled GJ anastomosis in robotic pancreatoduodenectomy (RPD) [5].

In this study, we present our surgical techniques of GJ using robot-sewn or stapled anastomosis in RPD. Furthermore, the outcomes between robot-sewn and stapled anastomoses were compared.

## 2. Materials and Methods

### 2.1. Patients and Patient Selection

Forty consecutive patients who underwent RPD at our institution between September 2020 and October 2022 were included in this study. The following clinical data of the included patients were extracted from the prospectively collected database: age, sex (male or female), body mass index (BMI), American Society of Anesthesiologists (ASA) physical status [6], comorbidity (hypertension and diabetes mellitus), primary disease (cancers or others), operative time, estimated blood loss, anastomosis time of GJ, mortality, postoperative complication, DGE, anastomotic leak at GJ, postoperative bleeding from GJ, postoperative pancreatic fistula (POPF), and hospital stay. Postoperative complications which occurred within one month after surgery were collected and graded based on the Clavien–Dindo classification [7]. Complications with the Clavien grade ≥ 3 were regarded as major complications. DGE and POPF were assessed according to the International Study Group of Pancreatic Surgery (ISGPS) [8,9].

Regarding the indication for RPD, there were no contraindications for patient’s age, BMI, or previous abdominal surgery; however, the initial indication included selected patients with benign and low-grade malignant diseases instead of advanced tumors requiring vascular resections [10].

### 2.2. Surgical Technique

The robotic platform was performed using a daVinci Si or Xi system (Intuitive Surgical, Sunnyvale, CA, USA). Our surgical protocol and strategies for RPD based on a multicenter, nationwide training program in the Netherlands (LAELAPS-3) was previously reported [10,11,12]. Briefly, the resection commenced with the extended Kocher’s maneuver. The subtotal stomach-preserving technique was used. Following the dissection of the hepatoduodenal ligament, the pancreatic neck was divided; the division was followed by uncinate dissection. The reconstruction started with pancreaticojejunostomy using the modified Blumgart method. Subsequently, hepaticojejunostomy anastomosis was performed. Finally, antecolic GJ anastomosis was performed using intra-abdominal robot-sewn or stapled methods.

In this study, all anastomoses were performed by a single console surgeon (KT) who obtained sufficient experiences of both robot-sewn and stapled GJ anastomosis through the LAELAPS-3. Our initial protocol for GJ surgery included robot-sewn anastomosis, which was changed to stapled anastomosis after 2022.

#### 2.2.1. Robot-Sewn Gastrojejunostomy

Robot-sewn GJ anastomosis was performed using the Albert–Lembert suture. Initially, the posterior seromuscular layer between the stomach and jejunum was sutured using running 3-0 V-loc sutures. Next, the ends of the stomach and jejunum were opened and anastomosed with running 3-0 V-loc sutures from the posterior to the anterior side. Finally, an anterior anastomosis of the seromuscular layer was performed. An overview of the GJ robot-sewn anastomosis is shown in Figure 1 and Appendix A.

#### 2.2.2. Stapled Gastrojejunostomy

The standard protocol for stapled anastomosis includes side-to-side GJ anastomosis between the posterior side of the stomach and jejunum (Appendix A). After the small holes in the stomach and jejunum were made, a 60 mm stapler was inserted and stapled (Figure 2). Careful and precise hemostasis of the stapled line should be confirmed from the entry hole. Thereafter, the entry hole was closed using running 3-0 V-loc sutures.

### 2.3. Postoperative Protocol after Surgery

Our standard protocol after surgery was as follows: early scheduled mobilization started on postoperative day (POD) 1; nasogastric tube removal started on POD 2-3; oral intake started on POD 2-3; early removal of urinary catheter started on POD 3; and early removal of drains started within 7 days after surgery. All patients were managed with the same protocol.

### 2.4. Statistical Analysis

Data are presented as proportions for categorical variables and mean with standard deviation (SD) for continuous variables. Statistical differences between groups were assessed using Fisher’s exact test or chi-square test for categorical variables, and the Mann–Whitney U-test for continuous variables. JMP version 11 software (SAS Institute, Cary, NC, USA) was used for statistical analysis.

## 3. Results

### 3.1. Patient Characteristics

Demographic and clinical outcomes of the 40 patients (27 male and 13 female) are shown in Table 1. Malignant diseases were observed in 22 patients, including 5 with pancreatic cancer. Of the 40 patients, 19 had robot-sewn GJ, and 21 had stapled GJ anastomoses. In terms of comparison between robot-sewn and stapled GJ, no significant differences were found in preoperative factors, including comorbidities and primary diseases.

### 3.2. Surgical Outcomes

Intraoperative outcomes included the mean (SD) operative and GJ times of 428 (63.5) and 34.0 (15.0) minutes, respectively. The overall incidences of DGE (grades B and C) and POPF (grade B) were 10.0% and 7.5%, respectively. Neither anastomotic leak at GJ nor postoperative bleeding from GJ were confirmed. The mean (SD) postoperative hospital stays were 11.6 (5.3) days.

Outcomes between robot-sewn and stapled GJ are demonstrated in Table 2. Regarding intraoperative outcomes, the stapled group had a significantly shorter GJ time than the robot-sewn group (22.7 min versus 46.5 min, *p* < 0.001). Furthermore, stapled GJ cases were significantly associated with a lower incidence of DGE (0% versus 21%, *p* = 0.01). Although not significant, the stapled group tended to have shorter postoperative hospital stays (9.9 days versus 13.5 days, *p* = 0.08).

## 4. Discussion

The present study describes our surgical techniques for GJ using robot-sewn or stapled anastomosis in RPD. Our standardized protocol for surgical techniques for GJ should help to understand its tips, tricks, and pitfalls and help to introduce RPD safely. Moreover, the outcomes between robot-sewn and stapled anastomoses during RPD were investigated. We found that stapled GJ anastomosis could be associated with shorter GJ time as well as a lower incidence of DGE than robot-sewn anastomosis.

A previous study from the University of Pittsburgh Medical Center group has reported that a longer length of the GJ anastomosis and a robot-sewn anastomosis were associated with decreased DGE in RPD [5]. Although their findings were contrary to our results, a previous meta-analysis suggested that the stapled GJ anastomosis may be associated with a lower incidence of DGE in open PD, without increasing other complications such as POPF and anastomotic leak [4]. In addition, another previous meta-analysis reported that several clinical factors, including diabetes mellitus, POPF, and postoperative complications, were risk factors for DGE [13]. Therefore, the optimal anastomotic method for GJ remains under debate, especially in RPD.

With regard to outcomes between robot-sewn and stapled GJ, we suggested that a longer length of GJ anastomosis using a 60 mm stapler instead of a 45 mm stapler might lead to lower incidence of DGE in the stapled GJ group [5]. However, accurate length of both GJ anastomoses was not measured in this study. Moreover, anastomotic edema during robot-sewn GJ might be related to the development of DGE. Further studies should be conducted to investigate the predictors associated with DGE as well as long-term outcomes following RPD.

The possible advantages and disadvantages of robot-sewn and stapled GJ anastomoses in RPD are summarized in Table 3. Although robot-sewn GJ anastomosis requires a longer anastomosis time, it depends on the console surgeon’s skill. In addition, the robot-sewn GJ technique has no risk of stapler misfiring. Therefore, robot-sewn GJ anastomosis should be selected when an assistant has less experience in minimally invasive surgery. In contrast, stapled GJ anastomosis can be performed in a shorter period of time. However, it is difficult for an assistant to handle the stapler, and the technique depends on the assistant’s skills. A misfire by a stapler could lead to serious problems, such as conversion to open surgery or reoperation. Therefore, stapled GJ anastomosis should be performed after an assistant obtains experience with minimally invasive surgery. In contrast, extra-abdominal GJ is another option which is used in open surgery. Extra-abdominal GJ is technically easier and faster than intra-abdominal GJ; however, additional midline incision is required for GJ. Accordingly, it is important for robotic surgeons to understand the advantages and disadvantages of various GJ techniques and select the most suitable technique depending on their experience.

Considering the pros and cons of various GJ techniques, as shown in Table 3, we selected robot-sewn GJ anastomosis during our initial phase, although both robot-sewn and stapled GJ techniques were available for a console surgeon (KT). After obtaining adequate experience with robotic surgery as a team, the protocol was simply changed to stapled GJ anastomosis in 2022. The fact that the incidence of DGE by robot-sewn GJ anastomosis was approximately 20% also contributed to changing the protocol.

This study had several limitations. Although we demonstrated two major GJ methods in RPD, there are other options for GJ anastomosis. As we removed the specimen from the Pfannenstiel incision, our protocol included intra-abdominal GJ anastomosis using robot-sewn or stapled methods. However, extracorporeal GJ anastomosis is easier when the specimen is removed from the umbilical incision. Moreover, our protocol did not include the pylorus-preserving technique. Therefore, no investigation was performed to compare outcomes between GJ and duodenojejunostomy during RPD. Another concern is that our results were based on our limited experience with RPD. Considering the small number of patients included in this study and the nature of a retrospective study as opposed to that of a randomized controlled study, no definitive conclusions can be drawn as to which methods are preferable for GJ anastomosis in RPD. Although we have been trained through a nationwide training program in the Netherlands (LAELAPS-3) and successfully introduced the RPD program in Japan [10,11], the results might be affected by individual and institutional learning curves. Last but certainly not least, the present study has concerns as to whether this data can be applicable to a population with significantly higher BMI in the West.

## 5. Conclusions

We present our robot-sewn or stapled GJ technique for RPD. Surgeons should select a suitable method for GJ anastomosis based on their surgical method, as well as individual and institutional experiences with RPD. Our findings suggest that stapled GJ anastomosis might decrease anastomotic GJ time and incidence of DGE.

## Figures and Tables

**Figure 1 jcm-12-00732-f001:**
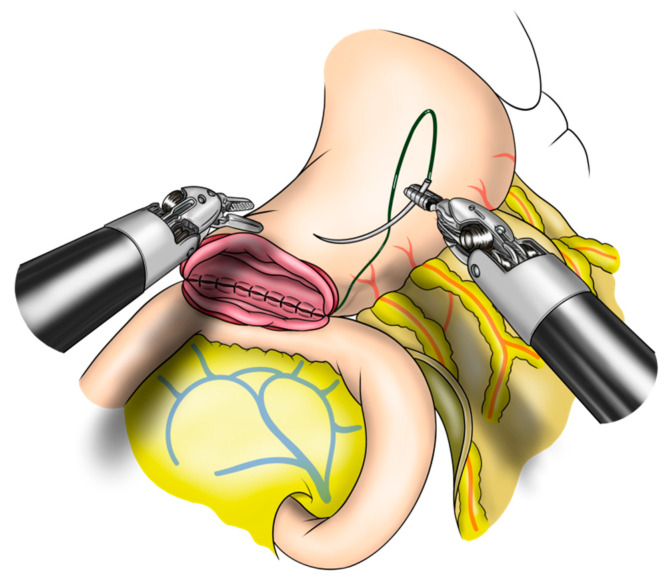
Robot-sewn gastrojejunostomy anastomosis in robotic pancreatoduodenectomy.

**Figure 2 jcm-12-00732-f002:**
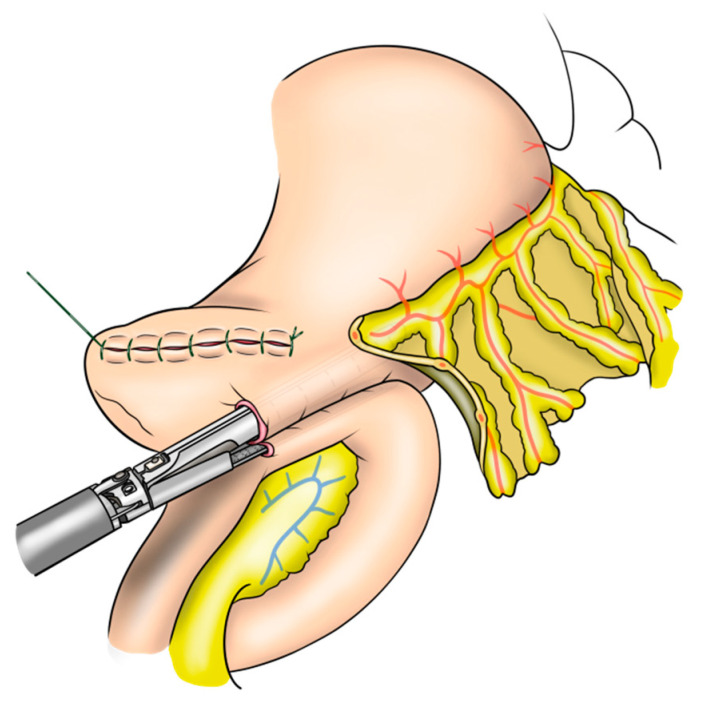
Stapled gastrojejunostomy anastomosis in robotic pancreatoduodenectomy.

**Table 1 jcm-12-00732-t001:** Characteristics between robot-sewn and stapled gastrojejunostomy anastomosis in robotic pancreatoduodenectomy.

Variables	Robot-Sewn Group	Stapled Group	*p* Value
No of patients	19	21	
Age, y	66.1 (11.6)	67.3 (13.8)	0.72
Gender			
Male	14 (74)	13 (62)	0.43
Female	5 (26)	8 (38)	
BMI, kg/m^2^	23.6 (3.6)	22.8 (2.3)	0.64
ASA physical status			
Grades 1–2	17 (89)	20 (95)	0.49
Grade 3	2 (11)	1 (5)	
Comorbidity			
Hypertension	8 (42)	5 (24)	0.22
Diabetes mellitus	2 (11)	4 (19)	0.45
Primary diseases			
Cancers	10 (53)	12 (57)	0.77
Ampullary carcinoma	5	2	
Duodenal cancer	3	3	
Bile duct cancer	1	3	
Pancreatic cancer	1	4	
Benign tumor	9 (47)	9 (43)	
IPMN	1	4	
Duodenal tumor	3	3	
Others	5	2	

Data are presented as mean (standard deviation—SD) or number (percentage). BMI—body mass index; ASA—American Society of Anesthesiologists; IPMN—intraductal papillary mucinous neoplasm.

**Table 2 jcm-12-00732-t002:** Outcomes between robot-sewn and stapled gastrojejunostomy anastomosis in robotic pancreatoduodenectomy.

Variables	Robot-Sewn Group (*n* = 19)	Stapled Group (*n* = 21)	*p* Value
Intraoperative factors			
Operative time, min	448 (68.2)	409 (54.0)	0.10
Estimated blood loss, mL	122 (116)	55.2 (80.3)	0.005
GJ time, min	46.5 (12.4)	22.7 (3.9)	<0.001
Conversion to open	0 (0)	0 (0)	-
Postoperative factors			
Mortality	0 (0)	0 (0)	-
Major complication (CD ≥ 3)	7 (37)	3 (14)	0.10
DGE (grades B and C)	4 (21)	0 (0)	0.01
Anastomotic leak at GJ	0 (0)	0 (0)	-
Postoperative bleeding from GJ	0 (0)	0 (0)	-
POPF (grade B)	1 (5)	2 (10)	0.61
Postoperative hospital stays, d	13.5 (6.8)	9.9 (2.8)	0.08

Data are presented as mean (standard deviation—SD) or number (percentage). GJ—gastrojejunostomy; CD—Clavien–Dindo classification [7]; DGE—delayed gastric emptying; POPF—postoperative pancreatic fistula.

**Table 3 jcm-12-00732-t003:** Advantages and disadvantages of various gastrojejunostomy anastomoses in robotic pancreatoduodenectomy.

	Advantages	Disadvantages
Extra-abdominal GJ	Familiar anastomosis in open surgery. Technically easier and faster compared to intra-abdominal GJ.	Additional midline incision is required for GJ. Need to undock the robotic system.
Intra-abdominal GJ	No need to undock the robotic system. Specimen can be removed from the Pfannenstiel incision.	Technically difficult compared to extra-abdominal GJ.
Robot-sewn GJ	Depending on the console surgeon’s skills. No risk of stapler misfiring.	Longer duration. Risk of anastomotic edema.
Stapled GJ	Shorter duration. Larger length of anastomosis.	Difficult handling. Depending on an assistant’s skills. Risk of stapler misfiring. Risk of anastomotic bleeding.

GJ—gastrojejunostomy.

## Data Availability

Not applicable.

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
