# Peer review of "Surgical Techniques of Gastrojejunostomy in Robotic Pancreatoduodenectomy: Robot-Sewn versus Stapled Gastrojejunostomy Anastomosis"

_jcm, 2023, doi:10.3390/jcm12020732_

Round 1
Reviewer 1 Report
Interesting article assessing robotic stapled v sewn gastroenterostomy and its influence on delayed gastric emptying after pancreatico duodenectomy.
several issues that need to be addressed
1. Almost half of the patients had a pancreaticoduodenctomy for non malignant conditions. The existence of malignancy or chronic pancreatitis or pre existing diabetes may influence DGE independent of the procedures. Although malignancy seems equally distributed, it is unclear whether there was any difference in spread of these benign conditions It would be useful to outline this spread to reassure the readers that this variable of prexisting conditions has not influenced the outcomes or at least to outline what were the reasons for the resection
2. It is unclear how the decision between stapled and handsewn was made. We are aware that these are consecutive patients but was there any influence by the operating surgeon based on preexisting or intraoperative decision making which led to decision to perform either procedure. Was it purely a one case stapled then next case handsewn etc. This would again exclude bias unconscious or otherwise in use of technique
3. The authors comment in their discussion that the length of the GE may influence DGE. although we are aware of the 60mm stapler, what was the length of the sewn gastroenterostomy. was it also 60mm or simply an ad hoc amount, and was there a standardised method of measuring the size of the sewn anastomosis
4. The authors comment that complications following resection can lead to DGE. However although they measure POPF, and GE bleeding, no other measures of complications are performed. Outlining even clavien dindo classification of complications to ensure that no significant infective, resp, cardiac or intra abdominal collections may be a variable to be considered in the results
5. In lines 109 to 113 under surgical outcomes, there are numbers given in parentheses. It is unclear what these numbers mean or refer to. I assume standard deviation but it should be clarified.
6. More widespread applicability of this data to a population with significantly higher BMI such as in Europe, subcontinent, australasia and north america should be mentioned as a limitation.
Author Response
January 10, 2023
Annie Du
Emily Tian
Special Issue Editor
Special Issue "Current Surgical Management of Pancreatic Cancer"
Journal of Clinical Medicine
Dear Editor:
RE: jcm-2149108
Surgical techniques of gastrojejunostomy in robotic pancreatoduodenectomy: Robot-sewn versus stapled gastrojejunostomy anastomosis
Thank you for reviewing our manuscript. We are pleased that our manuscript was favorably reviewed and found to be potentially acceptable for publication pending revisions.
We thank you and the reviewers for your thoughtful suggestions and insights. The manuscript has benefited from these insightful suggestions. I look forward to working with you and the reviewers to move this manuscript closer to publication in Journal of Clinical Medicine.
The manuscript has been rechecked and the necessary changes have been made in accordance with the reviewers’ suggestions. The responses to all comments have been prepared and attached herewith.
Thank you for your consideration. I look forward to hearing from you.
Sincerely,
On behalf of all coauthors,
Kosei Takagi, MD, PhD
Department of Gastroenterological Surgery, Okayama University Graduate School of Medicine, Dentistry, and Pharmaceutical Sciences, 2-5-1 Shikata-cho, Kita-ku, Okayama 700-8558, Japan
Tel: +81-86-223-7151; Fax: +81-86-221-8775; E-mail: kotakagi15@gmail.com
Reviewer 1
Interesting article assessing robotic stapled v sewn gastroenterostomy and its influence on delayed gastric emptying after pancreatico duodenectomy.
several issues that need to be addressed
Comment 1:
Almost half of the patients had a pancreaticoduodenctomy for non malignant conditions. The existence of malignancy or chronic pancreatitis or pre existing diabetes may influence DGE independent of the procedures. Although malignancy seems equally distributed, it is unclear whether there was any difference in spread of these benign conditions. It would be useful to outline this spread to reassure the readers that this variable of prexisting conditions has not influenced the outcomes or at least to outline what were the reasons for the resection.
Response:
Thank you for your feedback. Regarding the indication for RPD, there were no contraindications for patient’s age, BMI, or previous abdominal surgery; however, the initial indication included selected patients with benign and low-grade malignant diseases instead of advanced tumors requiring vascular resections. In the revised manuscript, we have described on this issue in the Materials and Methods (page 2, line 58-61). Furthermore, we have presented primary diseases more details especially for benign tumor in the Table 1. As shown in Table 1, there were no significant differences between the groups regarding the indication for RPD, including benign tumors.
Comment 2:
It is unclear how the decision between stapled and handsewn was made. We are aware that these are consecutive patients but was there any influence by the operating surgeon based on preexisting or intraoperative decision making which led to decision to perform either procedure. Was it purely a one case stapled then next case handsewn etc. This would again exclude bias unconscious or otherwise in use of technique.
Response:
Thank you for your feedback. As the reviewer 1 pointed out, this is an important issue that should be discussed. Considering the pros and cons of various GJ techniques as shown in Table 3, we selected robot-sewn GJ anastomosis during our initial phase although both robot-sewn and stapled GJ techniques were available for a console surgeon (KT). After obtaining adequate experience with robotic surgery as a team, the protocol was simply changed to stapled GJ anastomosis in 2022. The fact that the incidence of DGE by robot-sewn GJ anastomosis was approximately 20% also contributed to changing the protocol. In the revised manuscript, we have demonstrated on this issue more details in the Materials and Methods (page 2, line 72-75) and Discussion (page 6, line 179-184).
Comment 3:
The authors comment in their discussion that the length of the GE may influence DGE. although we are aware of the 60mm stapler, what was the length of the sewn gastroenterostomy. was it also 60mm or simply an ad hoc amount, and was there a standardised method of measuring the size of the sewn anastomosis.
Response:
Thank you for your feedback. A previous study from UPMC group has reported that a longer length of the GJ anastomosis, and a robot-sewn anastomosis were associated with decreased DGE in RPD (reference 5). In their study, they used either the 45-mm stapler or 60-mm stapler for the stapled GJ anastomosis. Therefore, we used a 60 mm stapler instead of a 45 mm stapler to create longer length of GJ anastomosis. We suggested that this might lead to lower incidence of DGE in the stapled group. However, accurate length of both GJ anastomoses was not measured in this study. Therefore, further studies should be conducted to investigate the predictors associated with DGE as well as long-term outcomes following RPD. In the revised manuscript, we have discussed on this issue (page 5, line 153-159).
Comment 4:
The authors comment that complications following resection can lead to DGE. However although they measure POPF, and GE bleeding, no other measures of complications are performed. Outlining even clavien dindo classification of complications to ensure that no significant infective, resp, cardiac or intra abdominal collections may be a variable to be considered in the results.
Response:
Thank you for your feedback. In the revised manuscript, we have added the data on postoperative complication within one month after surgery which were graded based on the Clavien-Dindo classification. Complications with Clavien grade ≥3 were regarded as major complications (page 2, line 51-56). As shown in Table 2, no significant difference was found in the incidence of major complications.
Comment 5:
In lines 109 to 113 under surgical outcomes, there are numbers given in parentheses. It is unclear what these numbers mean or refer to. I assume standard deviation but it should be clarified.
Response:
Thank you for your feedback. Data are presented as mean with standard deviation (SD) for continuous variables. In the revised manuscript, we have presented “mean (SD)” as the reviewer 1 suggested (page 4, line 120 and 123).
Comment 6:
More widespread applicability of this data to a population with significantly higher BMI such as in Europe, subcontinent, australasia and north america should be mentioned as a limitation.
Response:
Thank you for your feedback. We agree that this is an important limitation that should be mentioned. In the revised manuscript, we have described on this issue in the limitation (page 6, line 198-200).
Reviewer 2 Report
I have with great interest read this original article regarding stapled versus hand-sewn GJ during RPD. This is an interesting topic however, some questions arose when I read the manuscript.
1. I disagree a bit with the author’s interpretation regarding hand-sewn versus stapled anastomosis and think that to this date there is no clear evidence for either technique. That also goes for Gastrojejunostomy (GJ) versus Duodenojejunostomy (DJ). The author’s ref number 4 also includes PPPD. This should be clearer in the introduction as it is in the discussion.
However this does not lessen the role the data in this study provides.
2. From a technical view: Is this really a study of stapled versus hand-sewn anastomosis? Or is it a study regarding the placement of the anastomosis (the distal part of the stomach or backside of it)? One can argue for booth interoperations: this needs to be further explored in the discussion as well as that no conclusion in this study can be drawn regarding stapled GJ during RPD vs hand-sewn DJ during RPD
3. An obvious weakness of this study is that it is not randomized. As I understand the manuscript the technique was changed at the institution from hand-sewn to stapled and it is the result after this change that is reported. This is problematic, especially when reporting operating time. The difference might a you imply in the last sentence be explained with KT: s further training in RPD and not necessary the technique of the anastomosis. That also seems to be the explanation for lesser bleeding with stapling. Please explore this further in the discussion.
4. Since English is not my native language I have not done a complete review of the language, but I find the manuscript easy to follow and have no problems with the language.
Although there are some flaws in this study and some issues that needs to be clarified I still feel that the material with a revision can contribute to the field.
Regards
Author Response
January 10, 2023
Annie Du
Emily Tian
Special Issue Editor
Special Issue "Current Surgical Management of Pancreatic Cancer"
Journal of Clinical Medicine
Dear Editor:
RE: jcm-2149108
Surgical techniques of gastrojejunostomy in robotic pancreatoduodenectomy: Robot-sewn versus stapled gastrojejunostomy anastomosis
Thank you for reviewing our manuscript. We are pleased that our manuscript was favorably reviewed and found to be potentially acceptable for publication pending revisions.
We thank you and the reviewers for your thoughtful suggestions and insights. The manuscript has benefited from these insightful suggestions. I look forward to working with you and the reviewers to move this manuscript closer to publication in Journal of Clinical Medicine.
The manuscript has been rechecked and the necessary changes have been made in accordance with the reviewers’ suggestions. The responses to all comments have been prepared and attached herewith.
Thank you for your consideration. I look forward to hearing from you.
Sincerely,
On behalf of all coauthors,
Kosei Takagi, MD, PhD
Department of Gastroenterological Surgery, Okayama University Graduate School of Medicine, Dentistry, and Pharmaceutical Sciences, 2-5-1 Shikata-cho, Kita-ku, Okayama 700-8558, Japan
Tel: +81-86-223-7151; Fax: +81-86-221-8775; E-mail: kotakagi15@gmail.com
Reviewer 2
I have with great interest read this original article regarding stapled versus hand-sewn GJ during RPD. This is an interesting topic however, some questions arose when I read the manuscript.
Comment 1:
I disagree a bit with the author’s interpretation regarding hand-sewn versus stapled anastomosis and think that to this date there is no clear evidence for either technique. That also goes for Gastrojejunostomy (GJ) versus Duodenojejunostomy (DJ). The author’s ref number 4 also includes PPPD. This should be clearer in the introduction as it is in the discussion.
However this does not lessen the role the data in this study provides.
Response:
Thank you for your feedback. As the reviewer 2 pointed out, with regards to the optimal anastomotic method for GJ, controversy still exists on which anastomotic method is the best in open PD: stapled versus hand-sewn anastomosis; and GJ versus duodenojejunostomy. In the revised manuscript, we have changed the description in the Introduction (page 1, line 35-37).
Comment 2:
From a technical view: Is this really a study of stapled versus hand-sewn anastomosis? Or is it a study regarding the placement of the anastomosis (the distal part of the stomach or backside of it)? One can argue for booth interoperations: this needs to be further explored in the discussion as well as that no conclusion in this study can be drawn regarding stapled GJ during RPD vs hand-sewn DJ during RPD.
Response:
Thank you for your feedback. As we described the aim of this study in the Introduction, this is a study investigating the outcomes between robot-sewn and stapled GJ anastomoses (page 2, line 40-42). As the reviewer 2 suggested, we emphasized the aim of this study in the Discussion (page 5, line 141-142). Moreover, our protocol did not include the pylorus-preserving technique. Therefore, no investigation was performed to compare outcomes between GJ and duodenojejunostomy during RPD. This issue is a limitation in this study as shown in the revised manuscript (page 6, line 189-191).
Comment 3:
An obvious weakness of this study is that it is not randomized. As I understand the manuscript the technique was changed at the institution from hand-sewn to stapled and it is the result after this change that is reported. This is problematic, especially when reporting operating time. The difference might a you imply in the last sentence be explained with KT: s further training in RPD and not necessary the technique of the anastomosis. That also seems to be the explanation for lesser bleeding with stapling. Please explore this further in the discussion.
Response:
Thank you for your feedback. We agree with reviewer 2’s opinion that an obvious weakness of this study is that it is not randomized. In the revised manuscript, we have described this weakness of this study in our limitation (page 6, line 193-194). Moreover, we have revised the manuscript to demonstrate regarding the protocol for GJ anastomosis more details in the Materials and Methods (page 2, line 73-76) and Discussion (page 6, line 179-184).
Comment 4:
Since English is not my native language I have not done a complete review of the language, but I find the manuscript easy to follow and have no problems with the language.
Response:
Thank you for your feedback. The manuscript has been checked by Editage (www.editage.jp) for English language editing.
Round 2
Reviewer 1 Report
Authors have met my comments and clarified their paper to my satisfaction and i feel that it is suitable for publication